# Light Pollution and Cancer

**DOI:** 10.3390/ijms21249360

**Published:** 2020-12-08

**Authors:** William H. Walker, Jacob R. Bumgarner, James C. Walton, Jennifer A. Liu, O. Hecmarie Meléndez-Fernández, Randy J. Nelson, A. Courtney DeVries

**Affiliations:** 1Department of Neuroscience, Rockefeller Neuroscience Institute, West Virginia University, Morgantown, WV 26506, USA; jrbumgarner@mix.wvu.edu (J.R.B.); james.walton@hsc.wvu.edu (J.C.W.); jal0075@mix.wvu.edu (J.A.L.); ohm0001@mix.wvu.edu (O.H.M.-F.); randy.nelson@hsc.wvu.edu (R.J.N.); courtney.devries@hsc.wvu.edu (A.C.D.); 2Department of Medicine, Division of Oncology/Hematology, West Virginia University, Morgantown, WV 26506, USA; 3West Virginia University Cancer Institute, West Virginia University, Morgantown, WV 26506, USA

**Keywords:** light at night, cancer, circadian rhythms, clock genes, cell cycle

## Abstract

For many individuals in industrialized nations, the widespread adoption of electric lighting has dramatically affected the circadian organization of physiology and behavior. Although initially assumed to be innocuous, exposure to artificial light at night (ALAN) is associated with several disorders, including increased incidence of cancer, metabolic disorders, and mood disorders. Within this review, we present a brief overview of the molecular circadian clock system and the importance of maintaining fidelity to bright days and dark nights. We describe the interrelation between core clock genes and the cell cycle, as well as the contribution of clock genes to oncogenesis. Next, we review the clinical implications of disrupted circadian rhythms on cancer, followed by a section on the foundational science literature on the effects of light at night and cancer. Finally, we provide some strategies for mitigation of disrupted circadian rhythms to improve health.

## 1. Introduction

Overview of circadian rhythms: For the 3 to 4 billion years before electric light was invented, life on Earth evolved under the distinct pattern of light during the day and darkness at night. Along the way, nearly all organisms, including humans, internalized the temporal rhythm of Earth’s rotation and eventually developed self-sustaining biological clocks that align physiology and behavior to approximate the solar day. Indeed, virtually all biological processes display rhythms of function that approximate 24 h [1]. These so-called circadian (circa = about, dies = day) rhythms are endogenous, self-sustaining rhythms of optimal function that are set to precisely 24 h each day, primarily by exposure to light during the day. Well-known examples of mammalian circadian rhythms are the sleep-wake cycle, body temperature, and patterns of hormone secretion (e.g., cortisol and melatonin).

Importantly, several key processes involved in cancer are governed by circadian rhythms. For example, cell division shows strong daily cycles [2]. There is a bidirectional relationship between circadian rhythms and cell division. Disruption of circadian rhythms dramatically influences cell division and cancer development, whereas malignant transformation disrupts circadian organization [2]. As described below, common modern disruptors of circadian rhythms include night shift work, experiencing so-called ‘social jet lag’, which is shifted sleep-wake cycles between weekends and work days, and instances of exposure to light at night [3]. Below, we present a brief review of the molecular circadian clock system and the importance of light during the day and darkness at night. Then, we describe the role of circadian rhythms on the cell cycle, as well as the contribution of clock genes to oncogenesis. Next, we review the clinical implications of disrupted circadian rhythms on cancer, followed by a section on the foundational science literature on the effects of light at night and cancer. Finally, we provide some strategies for mitigation of disrupted circadian rhythms to improve health. Our search engines for the manuscript were PubMed and Google Scholar and our primary search terms were ‘light at night and cancer, light at night and circadian disruption, clock genes and cell cycle, and clock genes and cancer’.

The suprachiasmatic nuclei (SCN) of the hypothalamus is the master circadian clock in mammals [4]: it is positioned at the top of a hierarchy of independent endogenous time-keepers. The SCN is located directly above the optic chiasm and comprises ~50,000 densely packed small neurons in humans and ~20,000 neurons in rodents [5,6,7]. The cellular make-up of the SCN is diverse, and the SCN contains a variety of peptides and neurotransmitters [8,9]. 

As noted, the SCN serves as the master circadian clock at the top of a hierarchically organized system, however, circadian oscillators exist in virtually all tissues of multicellular organisms [10]. Tissue-specific clocks contain the molecular machinery necessary for self-sustaining rhythms [11] and have virtually the same molecular make-up as circadian oscillators in the SCN. Peripheral clocks are entrained to the external environment by the SCN via both neural and hormonal signals, as well as through non-SCN signals [12]. 

At a molecular level, the mammalian circadian system is driven by an autoregulatory feedback loop of transcriptional activators and repressors [10,13]. The proteins, circadian locomotor output cycles kaput (CLOCK), and brain and muscle arnt-like protein 1 (BMAL1) form heterodimers that promote expression of period (*Per1*, *Per2*, and *Per3*) and cryptochrome (*Cry1* and *Cry2*) via E-box enhancers [10,14,15]. Period (PER) and cryptochrome (CRY) proteins accumulate in the cytoplasm throughout the day. As the levels of these proteins increase, PER and CRY begin to form a heterodimer complex that translocates back to the nucleus to associate with CLOCK and BMAL1 to repress their own transcription [16,17]. This process requires about 24 h to complete a full cycle. In addition to the primary feedback loop, several additional regulatory loops influence the circadian clockwork, but the core clock components described above are protein products of clock genes that are essential for generation and regulation of circadian rhythms [18] Ablation of any of the core clock genes, Clock or Bmal1 [19], Per1 or Per2 [20], or Cry1 or Cry2 [21], disrupts circadian organization. Several secondary and tertiary clock components have been identified as necessary for the generation of precise circadian rhythms [22], and the criteria for what constitutes core clock genes are continuously evolving. For example, RevErb and Per3 were not initially considered critical for maintaining circadian clock function; however, the importance of these genes for circadian regulation is now widely accepted [23,24,25].

Light at night: Light is the most potent synchronizing factor for the SCN. In mammals, light information travels directly from intrinsically photosensitive melanopsin-containing retinal ganglion cells (ipRGC) in the eye through the retinohypothalamic tract to the SCN [26]. The SCN also receives indirect input from ipRGCs via the intergeniculate leaflet and input from rods and cones [27,28,29]. Indeed, photo entrainment persists in melanopsin-deficient mice but not in triple knockouts that lack melanopsin, rod, and cone function [30]. Once light information reaches the SCN, it signals via a multi-synaptic pathway to the pineal gland to regulate production and secretion of melatonin. Melatonin is secreted from the pineal gland at night in both nocturnal and diurnal animals. Circulating melatonin acts as a cue to entrain peripheral clocks via multiple interactions with the molecular clock mechanism, including phase-resetting clock genes [31]. Melatonin secretion is firmly controlled by light: <3 lux of light exposure at night is effective in shortening melatonin secretion duration and suppressing the onset of melatonin secretion in humans [32,33]. Due to ipRGC’s sensitivity to particular wavelengths, melatonin secretion is especially suppressed in short wavelength spectrum light [34]. Thus, artificial light at night has the potential to substantially alter physiology and behavior via suppression of melatonin rhythms.

At the molecular level, exposure to light rapidly induces Per1 [35,36]. A pulse of light during the dark phase can phase advance or delay the circadian clock depending on the timing, duration, and intensity of the light signal [37,38]. The circadian system also appears to be sensitive to light intensities below the threshold that induces phase shifts (e.g., [39]).

Exposure to dim light at night also influences the mammalian circadian system [40,41]. Several studies describe the effects of chronic and acute exposure to ecologically relevant levels of dim light at night (∼5 lux) on the circadian rhythms of rodent models [42,43,44,45]. Five lux of nighttime light exposure is comparable to levels of light pollution found in urban areas [46,47] and sleeping environments [48,49]. Exposure to chronic low levels of light at night alters circadian clock genes in both the SCN and peripheral tissues in mice and insects [50,51,52]. Exposure to dim light at night specifically attenuates the rhythm in Per1 and Per2 gene and protein expression in the SCN around the light/dark transition [50,52]. Furthermore, expression of *Bmal1*, *Per1*, *Per2*, *Cry1*, *Cry2*, and *Rev-Erb* are all repressed in the liver by exposure to dim light at night [50]. Nocturnal light may affect the liver through both autonomic and hormonal pathways [53]. Similar changes in circadian clock function are also apparent in the SCN of Siberian hamsters (*Phodopus sungorus*) exposed to dim light at night [54]. Hamsters exposed to 5 lux of light at night suppress PER1 and PER2 protein rhythms in the SCN independent of changes in locomotor activity rhythm. Importantly, rodents are typically more sensitive to lower-intensity lighting than humans [32,55]. However, light at night exposure also influences human circadian rhythms. The sleep patterns of people exposed to only natural light (outdoor camping) were more closely synchronized to solar time compared to individuals with exposure to light at night [56]. Furthermore, exposure to natural lighting reduced individual variability in melatonin and sleep rhythms, making late chronotypes more similar to early chronotypes [56].

## 2. Cell Cycle and the Circadian Clock

In normally proliferating mammalian cells, the rhythmic circadian clock and cell cycles are phase-locked [57]. The coupling of these two cycles remains an area of active research, however, several shared regulatory mechanisms highlight the molecular underpinnings of this coupling. First, it is important to understand how the cell cycle can regulate the circadian cycle, and vice versa.

Circadian clock regulation of the cell cycle generally occurs via interaction at the transcriptional or protein level between molecular circadian clock elements and [1] the cyclin-cyclin-dependent kinase (CDK) complexes and [2] the cell cycle inhibitors that regulate gating of the unidirectional progression through the stages of the cell cycle [58]. Initiation of the G1 phase of the cell cycle is regulated by c-Myc, which is transcriptionally regulated by CLOCK-BMAL1 [59]. Also controlling the transition to the G1 phase is the regulator of the Cyclin D-CDK4/6 complex, p16, which is rhythmically expressed under the control of the PER-NONO dimer [60]. ROR/REV-ERB regulates rhythmic p21 expression, which in turn regulates the expression of Cyclins A- and E-CDK2 and D-CDK4/6 complexes, which allow progression through G1 and S phases of the cell cycle [61,62]. Finally, the Cyclin B-CDK1 complex is regulated by WEE1, which is under circadian control by CLOCK-BMAL1 [63]. Taken together, transitions through all the stages of the cell cycle are under the control of the circadian clock. Conversely, evidence of cell cycle regulation of the circadian clock and rhythmicity is much more sparse, but compelling none the less, especially in the context of cancer. DNA damage can reset the circadian clock [64,65], which is potentially mediated by Cryptochrome [66]. Additionally, the tumor-suppressing gene p53 regulates Per2 expression by binding to the p53 response element, which in turn blocks binding of the CLOCK-BMAL dimer to the Per2 promoter [67].

In addition to reciprocal regulation between cyclic pathways, common enzymatic regulators in the phosphorylation and ubiquitination pathways are shared between the circadian clock and cell cycle pathways. These post-transcriptional modifiers could serve as both coupling and common regulatory mechanisms. Ubiquitination by the F-box protein FBXW7 leads to degradation of Cyclin E in the cell cycle and REV-ERBα and CRY2 in the circadian clock [68,69,70]. Degradation of PER1 and 2 in the clock and WEE1 in the cell cycle can be driven by ubiquitination by the E3 ubiquitin ligase β-TRCP [71,72,73], and degradation of PER2 and WEE1 can also involve phosphorylation via CK1δ [74,75]. Phosphorylation by GSK3β results in degradation of both Cyclin D1 and BMAL1, whereasREV-ERBα is stabilized [76,77,78]. Additionally, phosphorylation via AMPK results in stabilization of the cell cycle p27 protein [79], whereas the circadian clock protein CRY1 is destabilized [80]. Taken together, rhythms in the cell cycle and the circadian clock are normally coupled together, can reciprocally regulate each other, and have both shared and opposing effects by post-transcriptional modifiers. Dysregulation in the shared regulatory and coupling connections between the two pathways can be both necessary and sufficient for tumorigenesis.

## 3. Clock Genes and Oncogenesis

Because of the reciprocal regulation of core clock genes (CCG) and the cell cycle, it is not surprising that numerous studies establish an association between CCG and oncogenesis. Indeed, studies demonstrate a relationship between CCG and multiple cancer types, including breast, colorectal, endometrial, lung, prostate, pancreatic, and multiple lymphomas and leukemias [81,82]. Breast cancer patients frequently display mutations and increased methylation of the gene promoters in PER 1 and 2 [83,84,85]. Reduced expression of PERs and CRYs within breast tumors relative to surrounding normal breast tissue have also been reported [86,87,88,89]. Similarly, reductions in the expression of one or more PERs are seen in patient samples of colorectal, prostate, adrenal, ovarian, endometrial, lung, glioma, and pancreatic tumors [83,90,91,92,93,94,95]. In contrast, studies examining PER 2 expression in patient samples of gastric cancer report conflicting results. For example, Hu and Colleagues [96] report increased PER 2 and Cry1 expression in patient samples of gastric cancer, whereas Zhao and colleagues [97] report reduced expression of PER 1 and 2. Lower PER 1 and 2 expression is associated with shortened survival time. Reduced expression of PERs are associated with more severe tumor burden (PER 1) and shortened survival (PER 1, 2, 3) in non-small cell lung cancer [98]. Furthermore, low levels of PER 1, 2, and 3 expression in pancreatic cancer and PER 3 expression in colon cancer are associated with reduced survival [92,99].

In contrast to reduced expression of *PERs* and *CRYs*, *CLOCK* and *TIMELESS* expression are upregulated within breast tumors relative to surrounding normal breast tissue [84,87]. Increased *TIMELESS* expression in breast cancer patients is associated with reduced metastasis-free survival [100]. Increased expression of *CLOCK* or *NPAS2* prolonged metastasis-free survival in breast cancer patients [100,101]. *CLOCK* expression is significantly increased in patient samples of gliomas relative to the surrounding tissue [102]. Expression of *CLOCK* is significantly higher in high-grade gliomas relative to low-grade gliomas or normal brain [102]. In contrast, *CLOCK* expression is significantly reduced in ovarian and pancreatic cancer [92,93]. Reduced *CLOCK* expression within pancreatic tumors is associated with decreased survival time [92]. Clinical studies reporting associations between *BMAL1* and cancer are limited. *BMAL1* expression is significantly lower in patient samples of pancreatic cancer and head and neck squamous cell carcinoma [92,103,104]. Reduced BMAL1 expression in pancreatic tumors is associated with increased cancer severity and shortened patient survival [92,104]. Together, these findings emphasize the significant role of clock genes in cancer pathogenesis.

## 4. Disrupted Circadian Rhythms and Cancer (Clinical)

Since the late 20th century, the scientific community has considered disrupted circadian rhythms to be a potential risk factor for cancer development. Among other modern disruptors of circadian rhythms, the effects of artificial light at night (ALAN) exposure on tumorigenesis have been of particular interest because of its growing global presence. The disruptive effects of ALAN on endocrine function have led to a primary focus on the association between ALAN exposure and breast and prostate cancers. Because of the obvious ethical issues of causative ALAN and cancer studies, our understanding of ALAN exposure as a risk factor for cancer in humans primarily comes from epidemiological observational, case-control, and cohort studies.

The effects of ALAN on cancer development began to be considered in the late 1980′s and 1990′s [105]. Rodent [106] and in vitro [107,108] research around that time demonstrated beneficial effects of melatonin supplementation on tumor growth. This led to numerous hypotheses, including that suppression of melatonin rhythms via ALAN exposure leads to increased breast cancer incidence [109]. The first epidemiological studies to consider the effects of disrupted circadian rhythms on cancer development examined the health of shift-workers, such as nurses.

Shift-workers routinely experience multiple forms of circadian rhythm disruption, including mistimed eating, social jet lag, sleep deprivation, and ALAN exposure. This makes it difficult to parse out the disruptive aspects of shift-work that lead to harmful physiological consequences, such as increased cancer incidence. As a result, we must cautiously interpret the results of shift-work epidemiological studies in relation to the effects of ALAN, as most do not explicitly examine the effects of ALAN on physiology. But, the results of these studies in humans can provide an unfiltered view of how cancer incidence may be affected by circadian rhythm disruption resulting from shift-work and its behavioral consequences, including ALAN exposure.

Reports on the association between shift-work and breast cancer indicate the presence of increased risk with increased duration of shift-work in years. One of the first studies investigating shift-work examined breast cancer incidence in a Norwegian population of female telegraph workers over the age of 50. An increased odds ratio was observed but did not remain statistically significant after correction for duration of employment [110]. A subsequent study reported a significantly increased risk for the development of breast cancer in women who worked predominantly (>60%) night-shift jobs for more than half of the year [111]. Additionally, a case-control study of US female shift-workers in varying occupations reported that any prior history of graveyard shift-work was associated with increased risk for breast cancer, and this risk worsened with increased years of shift-work [112]. The same study suggested that delayed sleep also modestly increased the risk for breast cancer development [112].

Another set of case-control studies concluded that shift-work increased the risk of breast cancer in post-menopausal women working rotating night-shifts for more than 30 years [113] and in premenopausal women working rotating night-shifts for more than 20 years [114]. Similarly, a third case-control study of Norwegian female nurses reported that shift-work for more than 30-years increased the risk for breast cancer in comparison to non-shift workers, but no significant risk was observed in those working less than 30 years [115].

Other studies have presented null results on the effects of shift-work on breast cancer. For example, a German population-based case-control study described modest but non-significant effects of shift-work on breast cancer development [116]. Similarly, a cohort study of Chinese women detected no breast cancer risk associated with shift-work [117].

Shift-work is associated with an increased risk for several other types of cancer. A US population study of nurses living in 11 states reported that shift-work is modestly associated with an increased risk for endometrial cancer, but this risk was highest in women working rotating shifts for more than 20 years [118]. Rotating shift-work for more than 15 years is also associated with an increased risk for colorectal cancer in female nurses [119]. Men who work rotating day/night-shifts are also at increased risk for the development of prostate cancer in comparison to non-shift workers [120]. This finding was corroborated by a meta-analysis of eight additional shift-work and prostate cancer studies [121]. Notably, it is important to reiterate that shift-workers experience multiple forms of circadian rhythm disruption, including mistimed eating, social jet lag, sleep deprivation, and ALAN exposure. Thus, ALAN alone may not be as disruptive as shift work.

To provide a slightly more filtered view of the effects of ALAN on cancer incidence, others have considered rates of cancer in blind individuals [122]. ALAN exposure should not disrupt melatonin and other circadian rhythms in individuals lacking the functional retinal circuitry required to communicate light signals to the SCN and other components of the circadian system. Although this explicit hypothesis remains to be tested directly (i.e., blind shift-workers and cancer incidence, or correlation among blindness, ALAN exposure, and cancer), some studies have examined cancer incidence in blind individuals. The results are mixed. One study reported an overall reduction of cancer incidence in blind individuals, but individual risk ratios of breast and prostate cancers were not significantly different [123]. This reduction was only observed for ‘totally’ blind individuals, but not those with severe visual impairment. Another study comparing light-perceiving blind individuals to completely blind women concluded that the completely blind women had a lower risk for breast cancer [124]. Alternatively, a study of Norwegian women observed a non-significant association between breast cancer incidence and total blindness [125]. Contrary to the proposed hypothesis, a study of cancer incidence in the Finnish population reported that total blindness was positively associated with overall cancer incidence in males and with liver, stomach, and colorectal cancer in females [126]. The results of this study were challenged by Feychting and Ahlbom, who argue that the cohort size was too small, that incidence of diabetes may have explained some of the increased cancer risks, and that there is no overall coherency for cancer incidences between men and women [127]. Additional studies are needed to further test the associations among blindness, ALAN, and cancer incidence.

Epidemiological evidence to date suggests that there is a positive association between ALAN exposure and cancer in humans, but the results are not overwhelmingly supportive. The majority of epidemiological studies examined breast cancer risk ratios, while several others examined prostate and additional cancers. Environmental ALAN levels are associated with breast cancer incidence in humans. A positive association between breast cancer and ALAN exposure has been observed in several satellite imaging studies. Two studies of the Israeli population reported that urban environmental ALAN was associated with an increased risk for breast cancer [47,128]. Another satellite study concluded that Georgian women exposed to “high” levels of LAN (>41 watts per sterradian cm^2^) were at an increased risk for developing breast cancer in comparison to women exposed to “low” levels of ALAN (0–20 watts per sterradian cm^2^) [129]. A US population study reported a slight, but significant, association between ALAN levels and breast cancer incidence [130]. Another US study indicated that the women living in regions in the highest quintile of outdoor ALAN exposure were at an increased risk for breast cancer [131]. In contrast, one population-based case-control study concluded that residential outdoor lighting was not associated with breast cancer [132].

Two global ALAN and breast cancer incidence studies have been conducted. Using the GLOBOCAN 2002 database, Kloog and colleagues [133] described a significant association between environmental ALAN levels and breast cancer. In 2015, another group conducted a follow-up study and initially identified a weakened and non-significant relationship between ALAN and breast cancer incidence [134]. However, upon stratification of countries into Western, Gulf State and Southeast Asian, and “Other”, the association between ALAN and breast cancer incidence again was statistically significant [134].

Case-control and cohort studies examining ALAN and breast cancer are less conclusive than environmental studies. A survey study reported that bedroom ALAN levels were significantly associated with increased risk for breast cancer, although the study only collected bedroom lighting information from surveys that scored lighting levels between 0 (“completely dark”) to 4 (“very strong light”) [135]. A cohort study concluded that women who woke up and turned on lights during sleeping hours more than 2× a night at least 2× a week had an increased risk for breast cancer [136]. However, this same study reported no increased risk with lights being illuminated at night during sleeping hours when comparing higher versus lower frequencies [136]. Several studies have reported no association between breast cancer and indoor ALAN levels.

One previously discussed study of Californians reported no association between indoor lighting and breast cancer incidence [131]. Additionally, a case-control study of women in Connecticut, USA, did not determine any significant correlation between breast cancer and sleeping with the lights on or not drawing the curtains while sleeping at night [137]. Lastly, one study reported a negative relationship between nighttime waking with light exposure and premenopausal estrogen receptor-positive breast cancer in young women (in their 20’s; Hazard Ratio: 0.69, 95% confidence interval: 0.49–0.97) [138]. However, the most comprehensive meta-analysis to date examined 14 case-control and case-referential studies. Outdoor and indoor ALAN exposure was significantly associated with an increased risk for breast cancer [139].

Though breast cancer ALAN studies dominate association studies of ALAN and cancer, incidence rates of other cancers have been examined. Two previously discussed studies did not observe an association between ALAN levels and female incidences of lung [128], colorectal, liver, or larynx cancers [133]. Using satellite ALAN analysis, the same group identified an association between ALAN and increased prostate, but not lung or colon, cancer risk in men [140]. An observational study examining satellite ALAN data reported a significant correlation between blue-spectrum ALAN and prostate cancers when comparing the first and third tertiles of outdoor ALAN levels [141]. The same study noted a significant association between bedroom lighting levels and prostate cancer—this association was not observed in breast cancer [141]. In a subsequent follow-up study, Garcia-Saenz and colleagues [142] noted that broad-spectrum ALAN exposure was not associated with an increased risk for colorectal cancer, but ALAN in the spectral range of 425–560 nm (the authors call this a ‘blue’ index) was associated with an increased risk for colorectal cancer. Lastly, a global study examined effects of ecological light pollution by using the “protected area light pollution indicator” (PALI), which measures direct ALAN exposure, and the “protected area human influence indicator” (PAHI), which measures scattered ALAN exposure. A significant positive relationship was observed between PAHI and all forms of examined cancer and between PALI and colorectal cancer [143]. 

Together, current evidence supports an association between ALAN (and other circadian rhythm disruptors) and breast cancer. But, the relationship between ALAN and other cancers remains modest. Future studies designed to clarify the relationship between ALAN exposure and other forms of cancer should move beyond satellite imaging correlations. Although costly, the field instead might need to consider the use of wearable light-sensors that are capable of identifying ground-truth levels of ALAN exposure rather than relying on potentially misleading satellite data.

## 5. Disrupted Circadian Rhythms and Cancer (Foundational Science)

In contrast to the often conflicting clinical studies, compelling data exist for a causative effect of light at night on tumorigenesis in rodents. It is important to note that most laboratory rodents, such as mice and rats, are nocturnal in contrast to diurnal humans. In rodents, artificial light at night (ALAN) dysregulates clock gene rhythms [50,51,52], metabolism [144], immune function [145], hormone secretion [146], and reproductive function [42]. Notably, all of which intricately interact with cell growth and division. This has led foundational science research to investigate the role and relationship of artificial light at night as both a disruptor of circadian rhythms and facilitator of oncogenesis.

One of the first works that reported a positive relationship between light exposure and breast cancer was in 1964, when Jöchle noted that C3H-A mice had increased occurrence of spontaneous mammary tumors while exposed to constant light [147]. Subsequent studies demonstrated that Holtzman rats placed under continuous light exposure (LL) from birth and provided with 1,3-Dimethylbutylamine (DMBA) (10 mg/100 g BW), a carcinogen, had increased mammary tumorigenesis, elevated circulating prolactin, and higher DNA synthesis relative to rats housed in bright days and dark nights (LD) [148]. Administration of melatonin to these rats in a pattern that simulates nighttime exogenous release reversed the deleterious effects of constant light [148]. In a model of diethylnitrosamine-induced hepatocarcinogenesis, rats exposed to constant lighting had an increased number of and larger hepatocarcinoma nodules relative to the LD rats [149]. CBA female mice housed in constant lighting (24 h, 2500 lux) displayed irregular estrous cycle, increased incidence of spontaneous tumor formation (lung adenocarcinoma, leukemias, hepatocarcinomas), and shortened life span [150]. Male and female rats housed in LL conditions from one month of age demonstrated increased spontaneous tumorigenesis, accelerated aging, and decreased survival [151]. However, LL conditions which started at 14 months did not affect survival, and increased tumorigenesis only in females [151]. Further work from this laboratory demonstrated similar effects of constant illumination (2500 lux): rats housed in LL conditions had accelerated development of spontaneous tumors, development of metabolic syndrome, and accelerated mortality [152]. Constant lighting conditions are favorable for enhanced tumor growth. Indeed, 5 weeks of constant lighting increased blood glucose, reduced plasma triglycerides, and increased glioma tumor growth in rats [153]. Within the tumor microenvironment, LL conditions increased macrophage recruitment and upregulated genes involved in lipogenesis (increase in Acaca, Fasn, Pparγ, decrease in Srebp-1), glucose uptake (Glut-1), and tumor growth (Vegfα, Myc, Ir) [153]. LL conditions also enhance Wnt signaling pathways, thus increasing micro-vessels in the tumor and stroma and leading to hypervascularization [154]. Together, these studies demonstrate that constant lighting has a profound impact on tumorigenesis, tumor growth, and mortality.

Within the context of cancer, light at night research has recently shifted to the evaluation of ecologically relevant light levels, with the aim of modeling the nighttime light exposure common in the industrialized world. Sprague-Dawley rats exposed to dLAN (12 h light (300 lux):12 h dark (0.21 lux)) displayed higher rates of DMBA-induced mammary tumor growth, increased concentrations of serum estradiol, lower nocturnal 6-sulfatoxymelatonin secretion, and reduced survival compared to rats in dark nights [155]. Exposure to 0.2 lux of dLAN resulted in an accelerated rate of mammary cancer growth in female nude rats bearing human steroid receptor-negative MCF-7 breast tumors. LAN activated Akt stimulatory kinase phosphoinositide-dependent protein kinase 1 (PDK1), which correlated with an increased expression of proliferating cell nuclear antigen (PCNA) within tumors. In addition, circulating concentrations of IGF-1 were also significantly increased in rats exposed to ALAN [156]. Additional studies from the same lab have examined the relationship among ALAN, nocturnal melatonin, and tumorigenesis [157]. Human breast cancer xenograft-bearing rats were exposed to increasing intensities of ALAN, and results demonstrated a dose-dependent decrease in nocturnal melatonin levels, along with increased stimulation of tumor growth and mitogenic activity. To further elucidate the effect of melatonin concentrations on tumor growth, the researchers perfused rat xenografts with blood drawn from female volunteers during daytime (melatonin-deficient), nighttime (melatonin-rich), or nighttime after 90 min exposure to bright lights (melatonin-deficient). Xenografts perfused with melatonin-rich blood displayed suppressed proliferative activity compared to xenografts perfused with melatonin-deficient blood, thereby demonstrating that melatonin may be antioncogenic and that modulation of the hormone by light exposure is sufficient to alter proliferation [157]. Likewise, disruption of melatonin rhythms via exposure to ALAN lead to metastatic developments in the lung, liver, and brain of athymic nude rats with ERα+ MCF-7 breast cancer xenografts, whereas administration of exogenous melatonin reduced tumor development and metastatic lesions [158]. Moreover, exposure to artificial LAN promoted osteolytic bone metastases in MCF-7-bearing mice, and administration of melatonin during the inactive period reduced tumor burden in bone metastases, whereas luzindole (melatonin receptor antagonist) blocked the inhibitory effect [159]. 4T1 tumor-bearing BALB/c mice exposed to one 30 min period of ALAN (134 µ Wcm^−2^, 460 nm) increased tumor burden, reduced 6-sulfatoxymelatonin rhythms, and increased the number of lung metastases compared to LD [160]. These ALAN effects were once again diminished through administration of exogenous melatonin, hence reinforcing the hypothesis that circulating levels of melatonin can have a protective effect on tumor progression. Recent work has further explored the effects of light’s spectral composition on tumor growth. An inverse relationship between melatonin suppression and wavelength of light was demonstrated in 4T1 tumor-bearing BALB/c mice exposed to one of four different spectral compositions. Short wavelength light increased tumor growth, promoted lung metastases, and advanced DNA hypomethylation. Melatonin treatment, however, rescued these effects and reduced cancer burden [161]. Exposure to dim light at night can also have deleterious effects on cancer treatment. Indeed, studies demonstrate that ALAN promotes resistance to chemotherapeutics such as doxorubicin, paclitaxel, and hormonal therapy, namely, tamoxifen [162,163,164]. Together, these studies provide compelling data for a causative effect of light at night on tumorigenesis.

ALAN increases oncogenesis, but whether this increase is due to altered clock gene expression, reduced melatonin secretion, or likely combination of both remains to be determined. However, it is clear that there are beneficial effects of melatonin on tumor initiation and progression. Melatonin decreases the expression of matrix metalloproteinases and enhances the expression of adhesion proteins (i.e., β1 integrin and E-cadherin) [165,166], thus reducing cancer cell invasiveness and metastasis formation. Additionally, melatonin can decrease aromatase activity and reduce estrogen receptor alpha transcriptional action and expression [165,167,168], thereby reducing ERα+ breast cancer growth. Furthermore, melatonin can reduce tumor uptake of linoleic acid (LA) and prevent its metabolism to the mitogenic signaling molecule 13-hydroxyoctadecadienoic acid (13-HODE) [169]. Melatonin can also affect tumor growth indirectly via actions on immune cells. Specifically, melatonin can limit tumor growth by increasing natural killer cell activity [170] and counteracting tumor immune evasion by increasing IL-12, IL-2, and INF-γ production in T cells and monocytes, therefore driving T cells toward a Th1 response [171]. All the beneficial effects of melatonin on cancer initiation, progression, and immune cell function are mitigated by very low levels of light at night [172]. Indeed, melatonin secretion can be suppressed by exposure to as little as 0.03 lux at 480 nm in rodents and <3 lux in sensitive humans [33,173]. Given the numerous sources of ALAN [174], it is important to develop strategies to mitigate its oncogenic effects.

## 6. Conclusions

Several key processes involved in cancer growth are governed by circadian rhythms and there is a clear bidirectional relationship between circadian rhythms and cell division. Disruption of circadian rhythms influences oncogenesis. Indeed, clinical studies demonstrate a significant association between ALAN and breast cancer, and a modest relationship between ALAN and other cancers. Rodent studies provide compelling data for a causative effect of artificial light at night on tumorigenesis. In sum, current evidence supports an association between ALAN and oncogenesis (Figure 1). Given our current understanding of ALAN and tumorigenesis, there may be some strategies to mitigate the oncogenic effects of ALAN: (1) Turn off nighttime lights, (2) if nighttime lighting cannot be avoided, then wear short-wavelength (i.e., blue light) blocking glasses, and (3) in consultation with your physician, increase melatonin concentrations with melatonin supplements taken at night. 

## Figures and Tables

**Figure 1 ijms-21-09360-f001:**
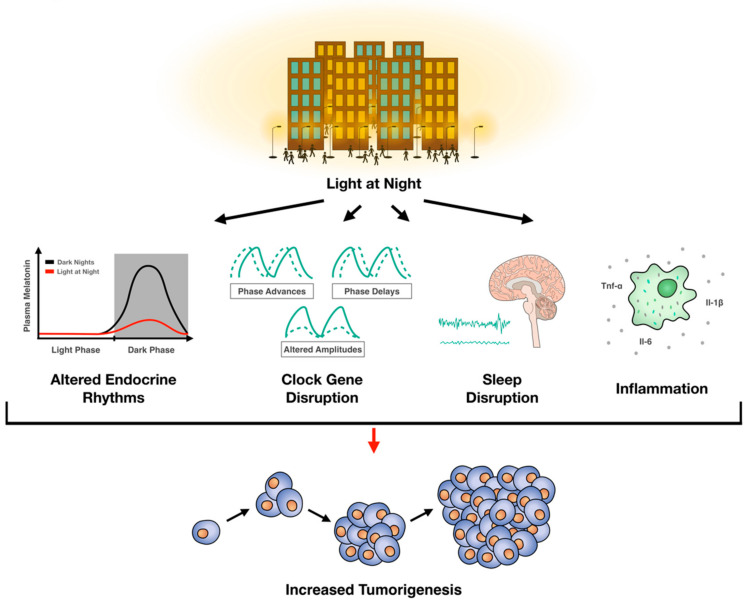
Summary of the effects of artificial light at night on tumorigenesis.

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
