# Peer review of "Light Pollution and Cancer"

_ijms, 2020, doi:10.3390/ijms21249360_

Round 1

Reviewer 1 Report

This review is targeting a highly relevant subject of the impact of artificial light at night on human cancer development. The subject is highly debated, especially the impact of outdoor illumination on the effects of circadian rhythm disruption. The manuscript provides a comprehensive summary of the status of the research and a lot of helpful literature.

However, the common thread is missing. First, the method of the review is missing, was it done systematically, if so which keywords were used and which search engines? Secondly, while some circadian clock systems are described in detail are other connections missing. Especially the subject melatonin is strongly underrepresented in this manuscript. Please refer to: Grubisic, M., Haim, A., Bhusal, P., Dominoni, D. M., Gabriel, K., Jechow, A., ... & Stebelova, K. (2019). Light pollution, circadian photoreception, and melatonin in vertebrates. Sustainability, 11(22), 6400. https://www.mdpi.com/573718

Circadian and seasonal disruption in mammals and many causes for cancer development and the reduction of anti-carcinogenic therapeutics is based on the reduction of melatonin. To my opinion, this is a central discussion point: How much artificial light at night can suppress melatonin and how much knowledge is available on the effects of artificial light at night in comparison to other factors, like noise, or psychological stress?

Please review the abbreviation LAN, light at night does include natural light such as moon and stars. Your manuscript, however, does not discuss literature on natural LAN. ALAN might thus be the better wording.

When discussing other organisms, especially the impacts on mice and rats it is important to mention that these are nocturnal. Please consider the discussion and maybe include some literature about day-active mammals if possible. The daytime light exposure may have a significant impact on cancer development in day-active organisms, including humans: See for example: Dauchy, R. T., Hoffman, A. E., Wren-Dail, M. A., Hanifin, J. P., Warfield, B., Brainard, G. C., ... & Dauchy, E. M. (2015). Daytime blue light enhances the nighttime circadian melatonin inhibition of human prostate cancer growth. Comparative medicine, 65(6), 473-485. https://www.ncbi.nlm.nih.gov/pmc/articles/PMC4681241/

The conclusion provides recommendations which are mainly based on the effects of suppressed melatonin e.g. avoidance of short-wavelength light at night time and taking melatonin as a supplement. However, without the introduction and discussion of the role of melatonin the reader is left alone, how the recommendations were drawn.

Please consider a discussion on the ALAN sources that can have an impact on cancer development, in light intensity and potential human exposure. These questions are of great concern to academia and the general public. Improving the manuscript in adding these discussion points will make it highly relevant.

Some remarks in detail:

  • Lines 78-84: Lacks description of how melanopsin receptors contribute to the circadian rhythm. Melatonin is suppressed by light especially in the short wavelength spectrum, in day-active as well as in nocturnal higher vertebrates.
  • Line 89: Bracket missing
  • Lines 95-97: ALAN also has an impact on clock genes in mosquitos: Honnen, A. C., Kypke, J. L., Hölker, F., & Monaghan, M. T. (2019). Artificial light at night influences clock-gene expression, activity, and fecundity in the mosquito Culex pipiens f. molestus. Sustainability, 11(22), 6220. https://www.mdpi.com/2071-1050/11/22/6220/htm
  • Conclusion: Wording “dramatically” how can the impact be dramatic when you listed studies with significant as well as insignificant results (Lines 278-294)

Reviewer 2 Report

This is a very detailed overview of the current state of knowledge in this area.  It’s strength is that is considers the limitation and potential confounders of many studies in a detailed manner.

I would suggest that a more clear distinction be made between night shift workers where a complete disruption and resetting of the circadian cycle occurs, involving a variety of neuroendocrine factors, and the nocturnal light pollution encountered in the modern world.  This latter state may not be as disruptive and shift work, but involves a large fraction of the population.

A summary figure might help to make the components of the review more readily accessible to the reader.

Author Response

Reviewer #2 (Comments to the Author): 

This is a very detailed overview of the current state of knowledge in this area.  It’s strength is that is considers the limitation and potential confounders of many studies in a detailed manner.

We thank the reviewer for their comments.

I would suggest that a more clear distinction be made between night shift workers where a complete disruption and resetting of the circadian cycle occurs, involving a variety of neuroendocrine factors, and the nocturnal light pollution encountered in the modern world.  This latter state may not be as disruptive and shift work, but involves a large fraction of the population.

To provide the suggested clarification, we have added additional details in L254-257.

A summary figure might help to make the components of the review more readily accessible to the reader.

As suggested, we have added a summary figure at the end of the manuscript. Initially, there was confusion on whether the graphical abstract was included within the manuscript.

Reviewer 3 Report

The present review by William H. Walker II addresses the effects of light exposure at nighttime on cancer promotion. Authors provide a comprehensive view on the current references and conclude that evidences support a role of light exposure at night on oncogenesis. They also inform on the tentative underlying mechanisms associating components of the circadian system (i.e. via the core clock genes and melatonin) with a role on cell proliferation, vascularization, etc. in tumorigenesis, and provide some strategies to mitigate these effects.

This is a well-organized, well-written and well-documented review, of interest to the readers. Thus, I recommend this paper for publication in its present form.

Some minor suggestions include:

L360: authors may want to separate with a space both parenthesis “) (“.

L416-417: What figure?

Author Response

Reviewer #3 (Comments to the Author): 

The present review by William H. Walker II addresses the effects of light exposure at nighttime on cancer promotion. Authors provide a comprehensive view on the current references and conclude that evidences support a role of light exposure at night on oncogenesis. They also inform on the tentative underlying mechanisms associating components of the circadian system (i.e. via the core clock genes and melatonin) with a role on cell proliferation, vascularization, etc. in tumorigenesis, and provide some strategies to mitigate these effects.

This is a well-organized, well-written and well-documented review, of interest to the readers. Thus, I recommend this paper for publication in its present form.

We thank the reviewer for their kind comments.

Some minor suggestions include:

L360: authors may want to separate with a space both parenthesis “) (“.

We have inserted a space on L360, now L380.

L416-417: What figure?

We have included the summary figure at the end of the manuscript. Initially, there was confusion on whether the graphical abstract was included within the manuscript.